# Modifiable Risk Factors of Non-Syndromic Orofacial Clefts: A Systematic Review

**DOI:** 10.3390/children9121846

**Published:** 2022-11-28

**Authors:** Angelo Michele Inchingolo, Maria Celeste Fatone, Giuseppina Malcangi, Pasquale Avantario, Fabio Piras, Assunta Patano, Chiara Di Pede, Anna Netti, Anna Maria Ciocia, Elisabetta De Ruvo, Fabio Viapiano, Giulia Palmieri, Merigrazia Campanelli, Antonio Mancini, Vito Settanni, Vincenzo Carpentiere, Grazia Marinelli, Giulia Latini, Biagio Rapone, Gianluca Martino Tartaglia, Ioana Roxana Bordea, Antonio Scarano, Felice Lorusso, Daniela Di Venere, Francesco Inchingolo, Alessio Danilo Inchingolo, Gianna Dipalma

**Affiliations:** 1Department of Interdisciplinary Medicine, University of Bari “Aldo Moro”, 70124 Bari, Italy; 2PTA Trani-ASL BT, Viale Padre Pio, 76125 Trani, Italy; 3UOC Maxillo-Facial Surgery and Dentistry, Department of Biomedical, Surgical and Dental Sciences, School of Dentistry, Fondazione IRCCS Ca’ Granda, Ospedale Maggiore Policlinico, University of Milan, 20100 Milan, Italy; 4Department of Oral Rehabilitation, Faculty of Dentistry, Iuliu Hațieganu University of Medicine and Pharmacy, 400012 Cluj-Napoca, Romania; 5Department of Innovative Technologies in Medicine and Dentistry, University of Chieti-Pescara, 66100 Chieti, Italy

**Keywords:** alcohol, cleft lip and palate, contaminants, drugs, orofacial clefts, pregnancy, prevention, pollutants, risk factors, smoke

## Abstract

OFCs (orofacial clefts) are among the most frequent congenital defects, but their etiology has yet to be clarified. OFCs affect different structures and functions with social, psychological and economic implications in children and their families. Identifying modifiable risk factors is mandatory to prevent the occurrence of non-syndromic OFCs (NSOFCs). PubMed, Cochrane Library, Scopus and Web of Science were searched from 1 January 2012 to 25 May 2022 and a total of 7668 publications were identified. Studies focusing on the risk factors of NSOFCs were selected, leading to 62 case-control and randomized clinical trials. Risk factors were categorized into non-modifiable and modifiable. The first group includes genetic polymorphisms, gender of the newborn, ethnicity, and familiarity. Within the second group, risk factors that can only be modified before conception (consanguinity, parental age at conception, socio-economical and educational level, area of residency and climate), and risk factors modifiable before and after conception (weight, nutritional state, acute and chronic diseases, psychophysical stress, licit and illicit drugs, alcohol, smoke, pollutants and contaminants) have been distinguished. This study provides a wide overview of the risk factors of NSOFCs, focusing on modifiable ones, to suggest new perspectives in education, prevention, medical interventions and clinical research.

## 1. Introduction

Orofacial clefts (OFCs) are characterized by clefts of varying widths that can affect the upper lip, the alveolar process, and the hard and/or soft palate [1]. Babies with OFCs manifest problems with phonation, feeding, hearing, and aesthetic and psychological discomfort. They require short- and long-term multidisciplinary health care [2]. In addition, these infants show higher mortality in the first years of life and up to 55 years of age and an increased risk of cancer [3].

Embryologically, the face is derived from five processes, namely the frontonasal and the two maxillary and mandibular processes [4]. The frontal process originates from the medial and lateral nasal processes bordering the bilateral nasal pits [5]. Between weeks 4 and 12 of intrauterine life, the development of the lip and palate occurs by migration into the first pharyngeal arch of cells derived from the neural crest. Following complex processes of cell differentiation, by fusion of the nasal placodes with the maxillary process, the upper lip and primary palate are formed. Subsequent elongation and thickening of the palatine processes result in their fusion and differentiation into bone and muscle tissue, creating the hard palate and soft palate [6]. Failure of fusion of maxillary and middle nasal processes results in a cleft lip (CL). Failure of fusion of the palatine processes results in the cleft palate (CP) [5,6].

According to the site affected, two main categories are distinguished: cleft lip with or without cleft palate (CL/P) and cleft palate (CP), mono or bilateral (Figure 1).

In addition, there is the subclass cheilognathoschisis (CL and alveolus) [6,7]. The most widely used classification system of OFCs sub-phenotypes is the LAHSHAL classification (Table 1) [8].

According to the aetiology, OFCs are classified into syndromic and non-syndromic. Syndromic OFCs (more than 400) originate from genetic, chromosomal, and teratogenic abnormalities and are associated with the presence or absence of other congenital and developmental and physical abnormalities. The most common genetic cause of OFCs is Van der Woude Syndrome, followed by Velocardiofacial Syndrome or 22q11.2 Deletion Syndrome and Pierre Robin Sequence and Associated Syndromes [3].

Non-syndromic OFCs (70% of CL/P and about 50% of CP cases), characterized by isolated morphologic manifestations, are determined by endogenous and exogenous etiologic factors [3,9,10]. 

The prevalence of OFCs varies according to race and geographic location: 1/500 among Asians and Native Americans; 1/1000 among Caucasian, Hispanic, and Latin populations; the lowest is found in the African people in the ratio of1/2500 [3,11].

Generally, OFCs are prevalent in the male sex. CL is present in 59.50% in males (M) and 40.5% of females (F); CP and CLP in 57.62% M and 42.38% F and 61.12% M vs. 38.58% F, respectively. In general, CP results in the most frequent (35.97%), followed by CLP (31.99%) and then CL (29.52%), submucosal schisis (SM) results in the least frequent (2.52%). CL is more frequent on the left and in males, in 61.2% M and 38.8% F, respectively [12,13]. In addition, 50% of patients with CL also present with CP due to a lack of fusion of facial extensions preceding palate formation [14].

Recent studies stated that the three types of OFCs (CL, CLP and CP) have different aetiology although the molecular dynamics remain unclear [15]. Scientific researchers have confirmed that different DNA methylation in embryonic development causes the occurrence of OFCs [16].

The analysis of blood samples from non-syndromic children with CL, CLP and CP revealed genomic sequences previously linked to OFCs and new (250) sequences, with different methylation occurring in the embryonic period [17]. These results support the hypothesis that the three subtypes may have different aetiology [3,18].

Besides data confirming the role of genetics in the aetiology of OFCs, scientific studies and epidemiological data show that environmental risk factors (e.g., smoking, alcohol and malnutrition in pregnancy, viral infections, teratogenic pharmacological agents, folate deficiency, weight) intervene significantly in the complex embryological development determining malformation [19,20,21,22,23,24,25,26,27,28,29,30,31,32,33,34,35,36,37,38,39,40,41,42,43,44,45].

During pregnancy, OFCs-related complications are commonly divided into maternal and fetal.

Maternal complications include miscarriage, elective termination of pregnancy due to prenatal identification of CAs, hyperemesis gravidarum, and anemia due to insufficient intake of folate and vitamin B12 [46,47,48].

Among fetal conditions, a study conducted in Brazil counts preterm birth (gestational age less than 37 weeks) before the stage of fetal weight gain, which is often associated with cesarean delivery [49]. A low-weight fetus (<1.5 kg) usually goes to post-partum mortality [50,51].

This systematic review aims to distinguish between non-modifiable and modifiable factors, focusing our attention on the latter, to take preventive, educational and therapeutic measures to reduce the occurrence of NSOFCs.

## 2. Materials and Methods

### 2.1. Protocol and Registration

The present systematic review has been performed in accordance with the recommendations of Literature search Preferred Reporting Items for Systematic Reviews and Meta-Analyses (PRISMA) and the International Prospective Register of Systematic Review Registry guidelines (PROSPERO) (ID: 341519).

### 2.2. Search Processing 

PubMed, Cochrane Library, Scopus and Web of Science were searched to find papers that matched our topic dating from 1 January 2012 up to 25 May 2022. The search strategy was built by using a combination of words that matched the purpose of our investigation, whose primary focus is the risk factors associated with OFCs. Hence, the following Boolean keywords were used: (“cleft” and “lip” and “palate”) (“orofacial” and “clefts”).

Separately, main gene polymorphisms associated with NSOFCs were searched and reviewed.

### 2.3. Inclusion Criteria

The articles were selected using the following inclusion criteria: (1) studies only on human subjects; (2) open access studies; (3) clinical trials, cohort studies and randomized controlled trials; and (4) English language.

### 2.4. Data Processing

Two reviewers (M.F. and A.P.) searched the databases to extrapolate the studies and assessed their quality independently, according to selection criteria. The selected articles were downloaded into Zotero (version 6.0.15). Any discrepancies between the two authors were resolved by consulting a senior reviewer (F.I.).

## 3. Results

### 3.1. Study Selection and Characteristics

A total of 7668 publications were identified, of which 4963 records using the key words “cleft lip and palate” and 2705 using the key words “orofacial clefts”. In the first case, the consulted databases were Pubmed (64), Scopus (2055), Cochrane (617), and Web of Science (2227); in the second case, the consulted database were Pubmed (9), Scopus (537), Cochrane (17), and Web of Science (2142). Hence, only studies that focus on the association between OFCs and risk factors were selected, by the analysis of the title and abstract. Studies dealing with other aspects of OFCs—for example, hereditary risk factors, associated diseases and syndromes, prenatal diagnosis, treatment and consequences—were excluded (7272), because they were off-topic. Duplicates (333) were removed, leading to 63 records selected, and after eligibility, 62 studies were selected for qualitative analysis (Figure 2).

### 3.2. Outcomes

The risk factors identified were classified into non-modifiable or modifiable. The first group includes genetic polymorphisms, gender of the newborn, ethnicity, and familiarity. Among the latter group, we made a distinction between (1) risk factors that can be prevented through educational programs before conception, namely, consanguinity, parental age at conception, socio-economical and educational level, area of residency and climate, and (2) risk factors modifiable both before and after conception, namely weight, nutritional state, acute and chronic diseases, psychophysical stress, licit and illicit drugs, alcohol, smoke, pollutants and contaminants.

## 4. Discussion

This systematic review aims to extrapolate from the literature of the last 10 years the main risk factors of NSOFCs, classifying them as non-modifiable and modifiable. Risk factors that can be partially modified by lifestyle interventions and/or educational programs before conception are included in the second group (Figure 3).

In NSOFCs, genetic polymorphisms belong predominantly to the folate pathway. Therefore, they will be treated in the paragraph of folate and vitamin deficiency, as their impact on embryogenesis becomes significant when an insufficient intake of folate and vitamin B12.

### 4.1. Non-Modifiable Risk Factors for Non-Syndromic Orofacial Clefts (NSOFCs)

#### 4.1.1. Gender of the Newborn and Ethnicity

Gender can be considered both a protective and a risk factor for OFCs. In some cases, studies identified correlations of gender with a specific type of OFCs. Specifically, several authors found that the male gender is a risk factor for CL/P and CL, while CP is more frequent in females [24,49,52,53]. According to Sakran et al., the male gender is generally a protective factor for CL/P [46].

Ethnic differences could potentially influence the incidence of OFCs. However, estimates of these relationships vary significantly throughout the research. This could be explained by differences in the measurement and categorization of ethnic groupings and variations in the classification of oral clefts [54]. The international literature identifies the Asian, white and indigenous races as having higher anomaly prevalence rates of OFCs. On the American and Asian Continents, 1/500 births are affected by this condition, compared to 1/2500 live births in African populations. Belonging to a non-white race could be a protective factor for the occurrence of OFCs [55]. These findings are also demonstrated by Shibukawa et al. in their cross-sectional study of Brazilian newborns from 2005 to 2016 [49]. The survey of Kapos et al. in agreement with the poor literature, showed that before United States-mandated folic acid supplementation, children born to American Indian mothers had a higher chance of developing NSOFCs than infants born to White mothers. Children born to Black mothers had a lower rate of this condition than infants born to White mothers, but it may be a result of differences in birth records’ methods for determining oral clefts [54].

#### 4.1.2. Familiarity

Family history is considered the most critical risk factor in the development of clefts [55].

The analysis of Silva et al. confirms earlier results of increased rates of familial recurrence due to a family history of CLP (about 40%) [55]. In addition, Sakran et al. reported positive family history as a significant risk factor for CLP in their cohort [46]. Ly at al. conducted an international study on paternal factors. They concluded that a father’s personal or family history of CLP was linked to a noticeably higher risk [56]. In the study of Noorollahian et al. a family history of this disease was present in about one-third of the cases [57].

The mechanism underlying this correlation is the likelihood of sharing alleles that are similar through descents [55].

### 4.2. Modifiable Risk Factors for Non-Syndromic Orofacial Clefts (NSOFCs) before Conception

#### 4.2.1. Consanguinity

Most genetic clefts have developed when consanguinity and family history have been discovered in a family simultaneously. About 9% of children worldwide are born due to consanguineous marriages, which account for 20% of the world’s population [58]. A case-control study by Sakran et al. on 600 patients assumed that parental consanguinity and family history were highly related to this condition [46]. Similarly, Cheshmi et al. concluded that the most notable factor that may have elevated the incidence of OFCs was consanguineous marriage [47]. In Iran, the prevalence of this custom is much more than the global medium due to cultural norms [47]. Compared to Western developed nations, where this sort of marriage is not particularly frequent, the high incidence of marriage in very traditional developing countries (such as Iran) greatly raises the risk of OFCs. These findings agree with other reports analyzed, which support the role of genetics in the incidence of oral clefts [59,60].

Silva et al. confirm that OFCs are inherited due to the consanguineous unions common in these areas. They also emphasize the need for genetic counseling for this community [55].

#### 4.2.2. Parental Age

Parental age acts as a risk factor and a protective factor for NSOFCs. Several studies agree that if both parents are ≥35 years or <19 years, the risk of OFCs increases [24,49,52,53]. According to de Carvalho et al., the risk increases for a paternal age ≥ 40 years [25]. Conversely, for other authors, parental age was not significantly associated with OFCs [46,56].

In regard to the potential protective effect of parental age, it is difficult to identify a range considered not at risk for OFCs because, in this field, the data in the literature are in contrast, especially about paternal age. If Sakran et al. confirmed that paternal age at childbirth of 25–29 years is a protective factor against CL/P, Olutavo et al. observed that paternal age ≥ 35 is surprisingly a protective factor for CL/P [23,46]. The same authors identified a maternal age between 26 and 35 years as protective [23]. In conclusion, these data reveal that protective age ranges have been identified for both parents (26–35 years and 25–29 years for mothers and fathers, respectively).

#### 4.2.3. Socioeconomic Level and Education

According to various research, fathers with lower socioeconomic positions and education levels are more likely to generate children with oral clefts [52,61].

A case-control study by Lupo et al. analyzed this association in a large population in Texas for about 10 years. They discovered that an unfavorable socioeconomic status is moderately related to oral clefts, particularly among Hispanics, using information from one of the world’s biggest active surveillance congenital disabilities registries. These results may significantly impact the prevention of health inequities [62].

The research of Figuireido et al. showed that poor parental education level appeared to be a risk factor for having a child with a cleft. Although this investigation did not find conclusive evidence linking this factor, the underlying cause may be related to prenatal care, nutrition, and other lifestyle choices [52].

#### 4.2.4. Area of Residency and Climate

Few research studies have looked at the connection between rural maternal residency and CLP risk, even though inequalities between rural and urban areas have been observed for numerous health and birth outcomes. Generally, moms who lived in rural areas were at a greater risk of having a baby with oral clefts [52,63,64,65,66].

Kapos et al.’s study evaluated maternal residence’s association with the risk of OFCs. In agreement with the poor literature, they showed that babies born to women who reside in rural regions have a greater chance of developing NSOFCs than babies born to moms who live in urban areas [54].

Noorollahian et al. found that most parents of OFCs-children reside in rural regions, leading researchers to conclude that this congenital defect is more common in people of lower socioeconomic standing [57].

An international study by Figuiereido et al. suggested a multifactorial etiology of OFCs, including environmental factors such as rural residency [52].

Teratogenic consequences of increased body core temperature have been demonstrated [21,67,68]. Soim et al. investigated the possible association between heat events of the weather and congenital malformations such as neural tube defects [69]. Despite research suggesting that warm temperature leads to worse pregnancy outcomes, authors found no statistically significant increased risk of neural tube malformations across all climatic regions. Future research should improve exposure and thoroughly review the topic [69].

### 4.3. Modifiable Risk Factors for Non-Syndromic Orofacial Clefts (NSOFCs) before and after Conception

#### 4.3.1. Weight

Obesity is a significant public health problem in developing countries [30,70]. Recent studies have considered the role of the mother’s body mass index (BMI) in the onset of pathologies affecting the fetus. BMI is considered normal if it is between 18.5 and 25 kg/m^2^. Maternal obesity can be responsible for congenital disabilities [71] as it can cause hyperglycemia, hyperinsulinemia, oxidative stress, systemic inflammation, and nutritional deficiencies that can compromise the development of the fetus [72,73]. Overweight mothers were found to have a higher risk of having babies with OFCs than women with normal BMI [30]. Kutbi et al. found that CP and CLP occurred more frequently in women who had higher than average BMI (≥35) in the pre-pregnancy period and that CL is not associated with changes in body weight [30]. Although previous revisions have found a 1.2 times greater risk in obese women giving birth to children with OFCs [71,74,75], Sato et al. in their prospective cohort study of 94,174 mothers, found no significant association between pathology and abnormal BMI (<18.5 and ≥25 kg/m^2^) [76]. Otherwise, few authors have shown a correlation between maternal underweight and OFCs [77,78]. George et al. analyzed the interaction between smoking and BMI. They found a strong negative influence only for CL and CP if the two factors are associated, especially in women with lower-than-normal BMI [79]. In this study, a reduction in the odds ratio for smoking was reported with increasing BMI, possibly for the ability of adipose tissue to retain the lipophilic chemicals of smoke [79,80]. However, a higher-than-normal BMI is not protective against the pathogenetic effects of smoking [79].

#### 4.3.2. Nutritional State

The maternal nutritional state is crucial to drastically reduce the appearance of congenital abnormalities (CAs) in newborns.

For instance, a strict maternal vegetarian diet increases 15 times the risk of OFCs in newborns when compared to omnivorous people [32]. Low levels of vitamin B12 and folate are often detected among vegetarian women, and a diet deficient in folate and vitamin B12, especially during early pregnancy, has been linked to OFCs [32]. One study showed a reduced amount of folate in the Indian population due to the type of vegetarian diet and mode of cooking food. Folate in vegetables is sensitive to heat, so it is destroyed during steaming [81].

Curiously, some studies deleted the relationship between cleft and specific foods, namely liver, sprue potatoes and caffeine-containing beverages. Caffeine belongs to the methylxanthine class and elevates the homocysteine level, interfering with the metabolic pathways of B6-vitamin [48]. Studies concerning liver consumption showed mixed results. According to some authors, it prevents the risk of CL/P because the liver is rich in nutrients [82]. Conversely, for others, preventing OFCs would not be sufficient [81]. Potatoes usually are considered a reliable source of nutrients. Still, when sprouted, they can contain high levels of glycoalkaloid solanine which is considered toxic and teratogenic for the human body. Periconceptional intake of this food may increase the risks of developing OFCs and neural tube defects [83].

#### 4.3.3. Psychophysical Stress

Another relevant environmental factor in modern times is stress.

A stressful event can cause an increase in cortisol in the maternal blood and therefore in the fetus. The teratogenic effect of corticosteroids (CTS) has long been demonstrated [84,85]. The enzyme 11B-hydroxysteroid dehydrogenase type 2 (11beta HSD2) responsible for regulating the passage of CTS through the placental barrier is down-regulated in case of stress [41,42]. Moreover, in the early stages of pregnancy—a critical moment for the formation of the facial massif—this enzyme is less represented [86]. The consequent increase in circulating CTS levels can also cause hyperinsulinemia and insulin resistance with negative effects on the development of the fetus [42,87,88].

It has been shown that taking vitamin B6 supplements helps reduce the harmful effects of CTS because this vitamin acts as a tissue receptor suppressor for these hormones [89].

Stress also increases catecholamines that reduce uterine blood flow and increase the risk of hypoxic damage to the fetus [84,90,91].

Ingstrup et al. demonstrated a strong correlation between the sudden death of a close relative or child in the prenatal period with the risk of developing cleft in the offspring [87].

In 2021, Sato et al. studied the psychological stress of mothers at 15 weeks of pregnancy through the Kessler Psychological Distress Scale (K6) [92,93] and concluded that there is an association with CL/P [76].

As a concerned maternal physical activity, its role has also been examined in the work environment. More strenuous movements such as twisting, lifting weights, running, and kneeling revealed a weak association with CL/P; however, longer sitting time proved to be a protective factor. However, further studies are needed to investigate further and confirm the correlation between maternal physical activity and OFCs [94].

#### 4.3.4. Acute and Chronic Diseases

##### Diabetes

Diabetes mellitus (DM) is a metabolic disease characterized by hyperglycemia. The regulation of blood glucose levels occurs thanks to insulin production by the pancreatic beta cells, a hormone that controls the entry and use of glucose into cells. Depending on the mechanisms responsible for the state of hyperglycemia, different forms of the pathology can be distinguished: (a) type 1 (or insulin-dependent DM), in which the pancreas does not produce insulin; (b) type 2 (or insulin-independent DM), in which the amount of insulin produced is insufficient to meet the body’s needs or does not act adequately on the tissues (insulin resistance); (c) gestational DM (GDM), in when increased blood glucose levels occur during the period of pregnancy. This condition resolves on its own after delivery, but affected women have a greater risk of developing type 2 DM in subsequent years.

Guariguata et al. reported in 2013 that the prevalence in the world of subjects affected by various forms of DM was 8.3% and that in 2035 this value would have doubled [95]. It has been seen that the association between GDM and obesity triples the risk for the fetus of developing CAs, including OFCs [26,27,31]. In 2016, Trindade-Suedam et al. conducted a study on the habits and health status of women who gave birth to children with OFCs, analyzing 325 cases observed during 12 months in a hospital in Sao Paulo (Brazil) dedicated to the management of these issues. The fasting glucose and abdominal circumference evaluation at the mother’s first prenatal examination were taken as reference values for diagnosing GDM. This study demonstrated that hyperglycemia during pregnancy increases the risk of incidence of OFCs in newborns [31].

Therefore, rigorous and systematic control of maternal blood glucose levels is necessary during the gestational period to prevent the risk of OFCs and other CAs [31].

##### Hypertension

In 2015, Figueiredo et al. confirmed what was already reported in 1995 by Hurst et al. who stated that a mother’s hypertension before pregnancy is associated with an increased risk of developing cleft, as is the use of antihypertensive drugs during the early stages of pregnancy which had been shown to have teratogenic effects on the fetus [31,52,96].

##### Angina Pectoris

The term “angina pectoris” comes from Latin and means chest pain, which is the main symptom of this disease. A temporary poor blood supply causes it in the heart, resulting in a lack of oxygen to the cardiac tissue. This condition is usually aggravated by physical exertion and emotional stress and is alleviated by taking nitro-glycerine [97].

Maternal angina pectoris (MAP) and myocardial infarction are ischemic events that rarely occur during pregnancy [98]. The study of Czeizel et al. was aimed at validating the correlation between MAP and OFCs reported in 2005 by Métneki et al. [99]. This study confirmed the greater predisposition of mothers affected by MAP to have children with OFCs compared to other CAs and shows that the drugs for the prevention/treatment of ischemic events have a low influence on the onset of the problem. Otherwise, the association of MAP with cigarette smoke increases the risk of developing clefts [29,100]. A genetic correlation between the two issues has also been hypothesized since hyperhomocysteinemia is associated with both a higher risk of MAP [101,102] and a higher incidence of OFCs [103], as well as genes related to stress [104]. However, the limitations of this study are linked to the low incidence of MAP in pregnant women and, therefore, to the small number of cases and controls [29].

##### Infections

Lip and palate development occurs during the early stages of pregnancy. In a recent study, Sakran et al. confirmed that exposure to infections during this phase increases the risk of developing NSCL/P [28,46]. Hyperthermia, often associated with viral infections, has been seen to play an important role in the onset of the cleft [81,105]. The most common conditions related to OFCs generally occur during early gestation and are the common cold, acute respiratory infections, influenza, pulpitis, cholecystitis, acute urinary tract infections, and pelvic inflammatory diseases [49]. These data suggest the importance of vaccines for pregnant women, such as the flu vaccine [81]. There is no evidence of a possible correlation between SARS-CoV-2 infection during pregnancy and the risk of OFCs.

##### Folate and Cobalamin Deficiency

Folate (vitamin b9) and cobalamin (vitamin b12) deficiency can be due to insufficient dietary intake and/or mutation in genes involved in the folate cycle, namely methylenetetrahydrofolate (MTHF) reductase gene, transcobalamin 2 (TCN2), and cystathionine beta-synthase (CBS) [106].

Although genetic polymorphisms are included among the non-modifiable risk factors, adequate folate and b12 supplementation in pregnancy compensate for enzyme deficiencies and the accumulation of teratogenic metabolites [106].

The enzyme MTHF reductase converts 5,10-MTHF to 5-MTHF, which is the active metabolite of folic acid and acts as a cofactor in several biochemical reactions, for example, the conversion of amino acid homocysteine to methionine, which lead to the methylation of DNA, RNA and histones proteins, one of the main mechanisms underlying gene expression regulation. Reduced levels of the active form of folic acid and accumulation of homocysteine impair cell differentiation and tissue growth during embryogenesis, leading to neural tube defects and OFCs. The common origin of these CAs is that the neural crest cells of orofacial tissues and teeth are situated in the dorsolateral regions of the neural tube. In addition, hyperhomocysteinemia is an independent risk factor for cerebrovascular disease associated with atherosclerosis, hypertension, inflammation, neurodegenerative diseases, pregnancy complications, and congenital malformation [107]. Even if it is clear that the supplementation with folate and cobalamin during pregnancy significantly reduce the prevalence of OFCs and neural tube defects both in mice and human [108], the correlation of OFCs with the most common polymorphisms of folate pathway, namely c.677C > T (rs1801133) and c.1298A > C (rs1801131), is still controversial. The recent Rai V meta-analysis demonstrated a strong association between C677T polymorphism and NSCL/P [109]. Jahanbin et al. found that only c.1298A > C polymorphism of the MTHFR gene may be a risk factor for NSCL/P in an Iranian population [110]. Instead, Komiyama et al. excluded the association of C677T and/or A1298C polymorphisms with the occurrence of CL/P in a large Japanese cohort. Recently, Machado et al. added new NSCL/P-associated genes of the folate/homocysteine pathway in Brazilian population, namely carbon pool by folate (hsa00670, *p* = 3.63 × 10^−11^), antifolate resistance (hsa01523, *p* = 0.00038), and biosynthesis of amino acids (hsa01230, *p* = 0.0460), formatted by (Formimidoyltransferase Cyclodeaminase) FTCD, (Methylenetetrahydrofolate Dehydrogenase, Cyclohydrolase And Formyltetrahydrofolate Synthetase 1) MTHFD1, Methylenetetrahydrofolate Reductase (MTHFR), 5-Methyltetrahydrofolate-Homocysteine Methyltransferase (MTR), Serine Hydroxymethyltransferase 1 (SHMT1), and Thymidylate Synthetase (TYMS) [111]. Conversely, according to Martinelli et al., the 677T allele of the MTHFR gene seemed to be protective against the occurrence of CL/P in the Asian population [112].

In addition, the same Italian group found the association of NSCL/P with polymorphism c.776C > G (p.Pro259Arg) and of the single nucleotide polymorphism rs4920037, belonging to transcobalamin 2 (TCN2) and cystathionine beta-synthase (CBS), respectively [113,114]. Therefore, it can be argued that NSCL/P also presents a genetic background interacting with environmental factors. The association of NSCL/P with specific polymorphisms might be influenced by the ethnic group and support the different genetic origin of CL/P and CL [114].

##### Other Diseases

In 2020, Ács et al. analyzed the correlation between children born with CP and any acute or chronic maternal illnesses during pregnancy [81]. This study found that the mother’s anemia may be linked to the pathology. This would agree with the theory that embryonic hypoxia, during the first three months of pregnancy, can hinder the correct formation of facial processes [81,115].

It has also been shown that the increase in inflammatory cytokines that occurs, for example, during pulpitis, can be responsible for the problem [81]. The same authors also point out that there is a higher risk of CP in the case of maternal hyperthyroidism in the first 14 weeks of gestation [81]. This finding agrees with studies in which Graves’ disease is linked to the development of congenital anomalies in the fetus [116].

Epilepsy also exposes the fetus to a greater risk of OFCs. Still, in these cases, the potential teratogenic effect of the antiepileptic drugs that the mother takes must also be taken into considered [81,117,118,119].

The children of women who suffered from migraines during pregnancy proved to be more predisposed to the development of clefts. Still, in this case, the influence of drugs taken to alleviate the symptoms must be investigated [81,120,121,122].

According to this article [81], even the spasmolytic and anti-pain drugs used for cholelithiasis, urolithiasis, and neuro-musculoskeletal pain syndromes can be identified as the main culprits of the problem [123].

#### 4.3.5. Licit Drugs and Vitamins 

Thirty-two percent of mothers used drugs during all the phases of pregnancy and, generally, several papers find a significant correlation between drugs and the occurrence of specific types of OFCs, especially during early gestation (Table 2) [31,46,53,124].

The most prescribed drugs in pregnancy are vitamins, followed by antibiotics. Current evidence is quite ambiguous regarding maternal supplementation with folic acid and multivitamins for preventing OFCs. According to most studies, the intake of folic acid or multivitamins containing folate at almost 400 mcg/day is mandatory to reduce the risk of neural tube defects and NSCL/P, not only in the first 3 months of pregnancy [33,46] but also during the periconceptional periods (from the time of conception to 12 weeks after conception) [48]. In other words, failing to take vitamins increases the risk of CL/P but not of CP [34,76]. This data may support the hypothesis that CL/P and CP appear to be genetically and embryologically distinct entities. In contrast, Roosendaal et al. found that periconceptional use of folate increases the risk of OFCs, especially of CL/alveolus, in a population from the Netherlands [35].

Specifically, the antibiotics most used are beta-lactams, followed by sulphonamides and macrolides. Antibiotics are prescribed more frequently in the second trimester, especially to women under 25 years [125]. Although the prescription of antibiotics in pregnancy has slightly increased in the last decade, the association between OFCs and antibiotics is still a matter of debate in the literature, also for molecules classically considered safe in pregnancy [125]. The exact pathogenetic mechanisms underlying the teratogenic effects of antibiotics have not yet been clarified. One of the best-known is trimethoprim inhibition of the folate methylation cycle [126]. According to a large cohort study, antibiotics use in early pregnancy does not represent a significant risk factor for NSOFCs, even if the risk increases for specific classes and selected periods. For instance, trimethoprim is associated with an elevated risk of CP during the first trimester; doxycycline/tetracycline and sulfamethoxazole are strongly associated with CL/P during the 2nd month, and pivmecillinam is associated with a high risk of CP during the 3rd month [125].

Controversial data concern amoxicillin, the antibiotic most used to treat respiratory and urinary infections in pregnancy and classified in the pregnancy category B drug by the USFDA (U.S. Food and Drug Administration). According to an analysis of the Slone Epidemiology Centre Birth Defects Study on 1348 infants affected by OFCs, maternal assumption of amoxicillin is associated with an increased risk of CL/P in the first trimester. Conversely, in the third trimester, amoxicillin is associated with an increased risk both of CL/P and CP [127].

CTS are widely used as anti-inflammatory and immune-modulating drugs during pregnancy and induce pulmonary maturity in the fetus. Above all, they are widely used during pregnancy in dermatological formulations for treating rashes, psoriasis, dermatitis and eczema, and in inhalation formulations to treat rhinitis [128].

Since 1951, the administration of oral CTS at high doses during early pregnancy has been considered a risk factor of OFCs because of teratogenic effects, which reach the acme at 60 mg/kg/day (dose of prednisone or equivalent) [128]. The increase in endogenous CTS can interfere with the fusion of the palatal shelves in mice. Considering that the fetus has a lower endogenous level of CTS and that the enzyme 11beta HSD2 is less represented in the early stage of pregnancy, it can be argued that even low doses of CTS and dermatological formulations, such as betamethasone, can overcome placenta, reach fetal circulation and cause teratogenic effects during embryogenesis [86]. As already discussed above, the level of endogenous cortisol produced in excess in case of maternal stress can also contribute to the risk of OFCs [87,129].

The first result from the National Birth Defect Prevention Studies (NBDPS), based on data from 1997 to 2002, found a correlation with CL/P but not CP [129]. Conversely, the updated data analysis from the same registry does not support the correlation between CTS and CL/P. Authors adjourned previous results with data from 2003 to 2009 concerning 2372 Newborns with CL/P. According to new data, there is a slight association between any use of all formulations of CTS and systemic CTS and CLP [128].

Among oral CTS, prednisone has shown a mild association with CL/P. Regarding nasal spray/inhaled formulations, the type of CTS with the stronger association is triamcinolone with CLP, followed by beclomethasone, which is associated with CLP and CP. No association was found with topic CTS [128]. In contrast to this assumption, the same author found the suggestion of an association of dermatologic CTS with both CLP and CP [128].

Regarding the timing of exposure, the most critical period of exposure to CTS is during weeks 1–4 and 5–8 after conception, followed by exposure only during 4 weeks before conception and exposure during weeks 5–8 and 9–12 after conception. The other periods are not considered at risk of CL/P. In conclusion, the use of CTS during pregnancy is not associated with the development of CL/P in the offspring, except for little evidence of associations between specific molecules, namely prednisone, triamcinolone, beclomethasone or timing, especially during weeks 1–4 and 5–8 after conception [129].

Ondansetron is a 5-HT3 receptor antagonist commonly used to contrast hyperemesis pregnancy. The association of ondansetron with CL/P is still a subject of debate. According to the retrospective cohort study of Huybrechts et al. on 1,816,414 pregnant women, first-trimester exposure to ondansetron is associated with a small increased risk of OFCs (5 cases per 10,000 prenatally exposed livebirths). Specifically, the type of oral cleft found was CP. No high risk for CL or CLP has been reported [130].

Topiramate is an anticonvulsant drug commonly used to treat epilepsy, bipolar disorder and migraine also during pregnancy. The association of topiramate with CL/P is dose-dependent; in fact, the RR increased at a dose ≥ 200 mg/day, commonly used to treat epilepsy. Data are referred to a period from 3 months before conception through one month after delivery [131].

Finally, other drugs associated with OFCs are reported in the most recent literature, namely anticonvulsants, retinoic acid, analgesics, benzodiazepines, antidepressants, stimulants, and anti-hypertensive drugs, and drugs containing iron and folate [31,34,53].

**Table 2 children-09-01846-t002:** Licit drugs taken in pregnancy associated with specific type of OFCs [CP: cleft palate; CL/P cleft lip with or without palate; CPL: cleft lip and palate, OFCs: orofacial clefts].

Drug	Type of OFCs	Gestational Period	References
Trimethoprim	CP	1st trimester	[129]
Doxycycline/tetracycline	CL/P	2nd month	[129]
Pivmecillinam	CP	3rd month	[129]
Amoxicillin	CL/P CL/P, CP	1st trimester 3rd trimester	[127]
Prednisone	CL/P	1st trimester	[125]
Triamcinolone	CLP	1st trimester	[125]
Beclomethasone	CLP, CP	1st trimester	[125]
Ondansetron	CP	1st trimester	[130]
Topiramate	CL/P	1st trimester	[131]
Anticonvulsants	CL/P	Not reported	[34]
Retinoic acid	CL/P	Not reported	[34]

#### 4.3.6. Illicit Drugs

Opioid abuse during pregnancy is a high-risk habit for the emergence of various CAs, including OFCs. It increases the risk of OFCs associated with other craniofacial anomalies by nearly three times [31]. Comparison between infants with opioid exposure and the control group revealed a significant increase in all subtypes of OFCs (OR of CL/P, CP, CL and CLP) [132].

Prenatal opioid exposure and other drugs can have severe and immediate consequences for newborns. Neonatal Opiate Abstinence Syndrome (NOWS) affects approximately half of all infants exposed to antenatal drugs. Mullen et al. examined 11,599 live births in hospitals at West Virginia University’s tertiary care center for 4 years (2013 and 2017) to better understand the relationship between NOWS and CL/P. The prevalence of CL/P was found to be greater in newborns with NOWS (6.79 per 1000 live births) than in the general population (1.63 per 1000 live births) [124].

#### 4.3.7. Smoke and Alcohol 

Smoke is a risk factor that plays an essential role in the etiology of CL/P. Several authors stated that smoking during early pregnancy increases the risk of CL/P by approximately 14.6 times in newborns [37,38].

Maternal exposure to numerous chemicals that are released into the air as a result of the incomplete combustion of tobacco and other organic compounds such as cooking and heating fuels have been shown to cause cracks; these substances include polycyclic aromatic hydrocarbons (PAHs), carbon monoxide (CO) and heavy metals [133].

Smoking increases levels of carbon monoxide, which has a high affinity for hemoglobin, thus resulting in reduced oxygen supply to the placenta. At the same time, nicotine constricts the uterine wall causing hypoxia in embryogenetic tissues during the palate genesis. Hypoxia induces malformations in the maxillary region [38].

Several researchers, including Shaw and Romitti 1996–1999, have shown that cigarette smoke can alter the expression of the gene involved in palatogenesis, transforming growth factor alfa (TFGA) [37,134,135]. Krapels et al. in 2004 showed that the risk of having a child with NSCL/P is increased by a factor of 3 in the case of smoking mothers with a mutated genotype compared to non-smoking mothers with at least one functioning allele [38,136].

The biological mechanisms underlying cleft lip and palate are hidden in the DNA and depend on a defect in the TFGA gene, which is responsible for organizing the structure of the face and palate during embryogenetic processes; the mechanism of interaction between smoke and gene remains unknown [134,135]. TFGA is responsible for promoting the synthesis of the extracellular matrix and the migration of mesenchymal cells and ensuring palate strength and fusion. When normal, this portion of DNA is insensitive to the negative effect of smoking, but when altered, it becomes vulnerable to smoking. If the fetus inherits the defect from both mother and father, it cannot detoxify itself and suffers the insult of cigarette smoke, which can result in malformation [134,135].

Regarding drug type, licit drugs such as tobacco, stimulants and antidepressants have been shown to increase the risk of orofacial malformations; this correlation was not observed for illicit drugs such as cocaine and cannabis due to the small sample size or lack of control [31].

Several authors have amply demonstrated the role of alcohol in the genesis of the palatal cleft. However, it varies with dose, frequency, and mechanisms of toxin transfer from the mother to the embryo [135,137,138].

Exaggerated alcohol consumption is associated with craniofacial malformations characterized by reduced growth of the central nervous system and neurodevelopment [138].

The author demonstrated an increased risk of partial or total CL/P in the first trimester of pregnancy compared to non-drinking women [139]. Between the fifth and tenth week of pregnancy is the period when the structures forming the palate and lip fuse. This is the most vulnerable period and alcohol abuse could affect the cells forming these structures with an increased risk of CL/P [140].

An important aspect relates to the amount of alcohol and frequency during pregnancy. One author has shown that consuming five or more glasses of alcohol in an evening can be particularly detrimental to fetal development because it exposes the fetus to a higher concentration of alcohol in the blood than someone who consumes the same amount over a more extended period. Another important aspect is genetic susceptibility, defined by maternal alcohol metabolizing genes (ADH1C haplotype gene) associated with reduced alcohol metabolism [138,139].

These authors’ limitations are the impossibility of quantifying exposure to these three causative factors, but this information can prevent and educate on the risk of orofacial malformations [140].

#### 4.3.8. Pollutants and Contaminants

Maternal exposure to gaseous air pollutants in the preconceptionally period and early stages of gestation revealed a positive correlation with the risk of developing NSOFCs. Several studies recognized an increased incidence of CL/P after exposure to CO (carbon monoxide), NO_2_ (nitrogen dioxide), and SO_2_ (sulfur dioxide). CP cases were related only to NO_2_ exposure [43,44]; no correlation was observed with O_3_ (ozone) [43,45]. During the 5–10 gestational weeks, exposure to environmental particulate pollution with a diameter ≤ 2.5 μm (PM2.5) is correlated with a 43% increase in CP risk for each 10 μg/m^3^ increase in PM2.5 concentrations [45]. The biological mechanism by which gaseous air pollutants can cause OFCs remains unclear. However, there are several hypotheses: hypoxia generated by CO; DNA methylation caused by NO_2_; and oxidative stress caused by SO_2_ [43].

Water pollution also proved a significant risk factor. Figueiredo et al. reported a higher incidence of OFCs in the offspring of mothers who drew water from a well than in controls who took filtered water or water drawn from public waterworks [56]. Nevertheless, water disinfection by-products (DBPs), which are formed by the interaction of chemical disinfectants and organic material in the water, are a hazard for OFCs’ occurrence. Exposure to DBPs or trihalomethanes (THMs) and haloacetic acids (HAAs), particularly HAA5, is correlated with the occurrence of CP [141].

Environmental contaminants include arsenic, pesticides, and heavy metals. Arsenic is widespread in the environment in organic and inorganic forms. Exposure to it can occur through occupational or dietary exposure, drinking water, or eating foods containing it, such as rice, vegetables, fruits, and shellfish. The study by Suhl et al. identified a positive correlation between CP and maternal occupational exposure to inorganic arsenic [142].

According to the same authors, there is a positive correlation between CP and paternal occupational exposure in the periconceptional stages to insecticide + herbicide + fungicide type pesticides [143].

However, the paternal influence in these malformations is controversial. According to Ly et al., there is no clear correlation between paternal exposure to environmental pollutants and the onset of OFCs [56].

During the second and third trimesters of pregnancy, in utero exposure to low concentrations of heavy metals, particularly mercury, lead, cadmium, and manganese, has not been proven to be a risk factor for CL/P [144,145].

Maternal occupational exposure in the periconceptional period to organic solvents (aromatic and chlorinated—which are widely found commercially in the form of degreasers, cosmetics, paints, detergents, or pesticides—has not shown a positive correlation with the occurrence of OFCs in offspring [145,146].

## 5. Conclusions

The aim of this systematic review was to demonstrate that NSOFCs recognize multiple modifiable risk factors and to highlight the interactions that genetics, environment, habits of daily life, diet and pathology have on the onset of this congenital abnormality.

Herein, risk factors were classified into non-modifiable or modifiable. The first group includes genetic polymorphisms, gender of the newborn, ethnicity and familiarity. Among the latter group, there are risk factors that can be prevented through educational programs and lifestyle interventions before conception aimed at planning a pregnancy safely, namely consanguinity, parental age at conception, socio-economical and educational level, area of residency, climate. Risk factors that are modifiable both before conception and during pregnancy include weight, nutritional state, psychophysical stress, acute and chronic diseases, licit and illicit drugs, alcohol, smoke, pollutants and contaminants.

The gender of the newborn is more associated with specific forms of OFCs, although conclusions are contradictory. Generally, CL/P is more frequent in males, while CP is prevalent in females. 

Caucasian, Asian, and indigenous ethnicities have higher rates of OFCs anomalies than non-white ethnicities. Familiarity is one of the major contributors to the occurrence of OFCs, especially if it coexists with consanguinity. Although OFC is a multifactorial disease, the genetic component has proven to be critical. This consideration stems from the high prevalence of OFCs in babies born from consanguineous relationships, which are uncommon in Western countries but common in less developed or developing countries (e.g., Iran).

More crucial is to understand the role of modifiable risk factors in reducing the incidence of OFCs in newborns. It is preferable to plan a pregnancy between 26–35 years for women and 25–29 years for men because, in these age ranges, the risk of OFCs is sensibly reduced, as well as to educate populations to avoid consanguineous marriages.

Few studies have examined the relationship between the area of residency and climate and the risk of OFCs, leading to the conclusion that even if there is a weak association between heat events and OFCs, the climate has no effect on the onset of this pathology. Conversely, rural residence is a risk factor compared to pregnant women’s residence in urban areas. This aspect, however, may be linked to another risk factor identified: socioeconomic status. A low level of socioeconomic development, which is common in rural areas and underdeveloped or developing countries, creates conditions that favor the onset of this pathology: consanguineous marriages, reduced access to care, insufficient intake of nutrients, abnormal BMI, unhealthy lifestyles and habits, exposure to pollution and contaminants.

Acute and chronic diseases, e.g., obesity, diabetes, hypertension, MAP, infections, folate deficiency and hyperhomocysteinemia, anemia, epilepsy, and hyperthyroidism have been linked to an increased risk of NSCL/P. Consequently, it is necessary to promote screening, vaccination, and health education plans to identify early pregnancies at risk and provide targeted medical or lifestyle interventions.

Nutritional deficiencies, particularly a lack of vitamin B12 and folate, are strongly linked to changes in neural tube defects, especially in carriers of the folate pathway genetic polymorphisms—a strict vegetarian diet carries a 15-fold increase in the risk of OFCs when compared to meat-consuming diets. In addition, some studies focused on the role of specific foods in pregnant women’s diets, namely sprue potatoes, liver and caffeine.

Psychophysical stress is a relevant environmental factor that is typical of modern times. This condition causes hormonal and biochemical changes in the pregnant woman, which may favor the onset of fetal development changes. It is preferable to treat psychological stress with non-pharmacological interventions because antidepressants are counted among the drugs associated with the risk of OFCs. Intense physical exercise has shown a weak association with the onset of OFCs. Although the protective role of moderate physical activity has been established, more research is needed to investigate and understand the relationship between maternal physical activity, psychophysical stress, and OFCs.

Drugs are widely used in pregnancy. Antibiotics, CTS, and specific drugs such as ondansetron and topiramate have received the most attention in literature, but their role is controversial. It is generally accepted that the intake of these drugs during the first trimester of pregnancy or before conception exposes the newborn to an increased risk of CAs, including OFCs. As well as the duration of intake, the dose of administration is also important, because high doses being linked to an increased risk of developing NSCL/P.

Licit or illicit use of opioids deserves special attention, because causes a threefold increase in the risk of developing OFCs and other craniofacial congenital anomalies. Furthermore, a link has been established between the onset of NSOFCs and the use and abuse during pregnancy of substances such as tobacco, antidepressants, and stimulants, whereas few studies have examined the role of illicit drugs such as cocaine and cannabis due to quantitative and ethical constraints in recruiting study samples.

Several studies have examined the effects of passive and active smoking on fetal development. Even though the recognized mechanisms of action differ, all studies agree that smoking and tobacco combustion products increase the risk of OFCs in infants by about 14.6-fold.

Alcohol consumption during pregnancy is strongly associated with OFCs, especially in cases of abuse or large intakes. It has been shown that consuming five or more glasses of alcohol in one evening can be harmful to fetal development. Most of the articles evaluated in the review identified smoking, alcohol, and drugs as major predisposing environmental factors for the development of NSOFCs, but many authors emphasized the limitations of studies on this interaction, which are typically due to the impossibility of quantifying exposure to these three causative factors.

Finally, periconceptional and early gestational maternal exposure to gaseous air pollutants, such as CO, NO_2_, and SO_2_, showed a positive correlation with the risk of developing CP. For every 10 g/m^3^ increase in PM2.5 concentrations, exposure to environmental pollution by particulate matter with a diameter of 2.5 m (PM2.5) was associated with a 43% increase in the risk of CP. In addition, water pollution (THMs, HAAs) and environmental contaminants (arsenic, pesticides) were also linked to OFCs. Conversely, an association with heavy metals or inorganic solvents, was not discovered.

Although multiple interactions and risk factors for the development of NSOFCs have been identified, the reviewers would like to highlight that the studies relate predominantly to mothers. The few studies that also take in consideration the paternal risk factor are referred to age, familiarity, area of residency, socioeconomic status, educational level, and environmental pollutants. Therefore, future research should focus on the paternal role, as well as on the mechanisms of action of environmental risk factors. At the same time, this review points out that most of the risk factors associated with cleft are preventable or modifiable. Educational programs, prevention campaigns, medical and lifestyle interventions are mandatory to obtain a reduction in the incidence of new cases of OFCs in the next years.

## Figures and Tables

**Figure 1 children-09-01846-f001:**
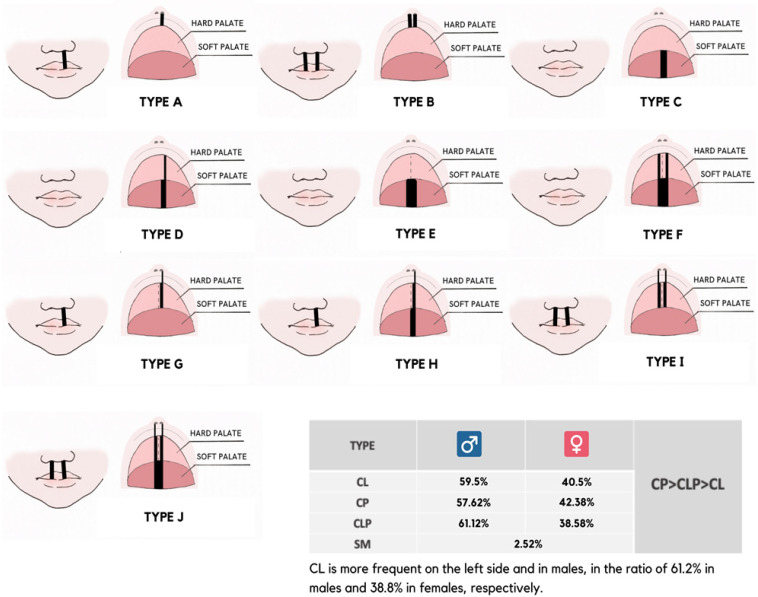
Anatomical classification of OFCs [CL: cleft lip; CLP: cleft lip and palate; CP: cleft palate]. Type A and type B: CL (mono and bilateral); type C and type E: soft CP (mono and bilateral); type D and type F: soft and hard CP (mono and bilateral); type G and type I: hard CLP (mono and bilateral); type H and type J: soft and hard CLP (mono and bilateral).

**Figure 2 children-09-01846-f002:**
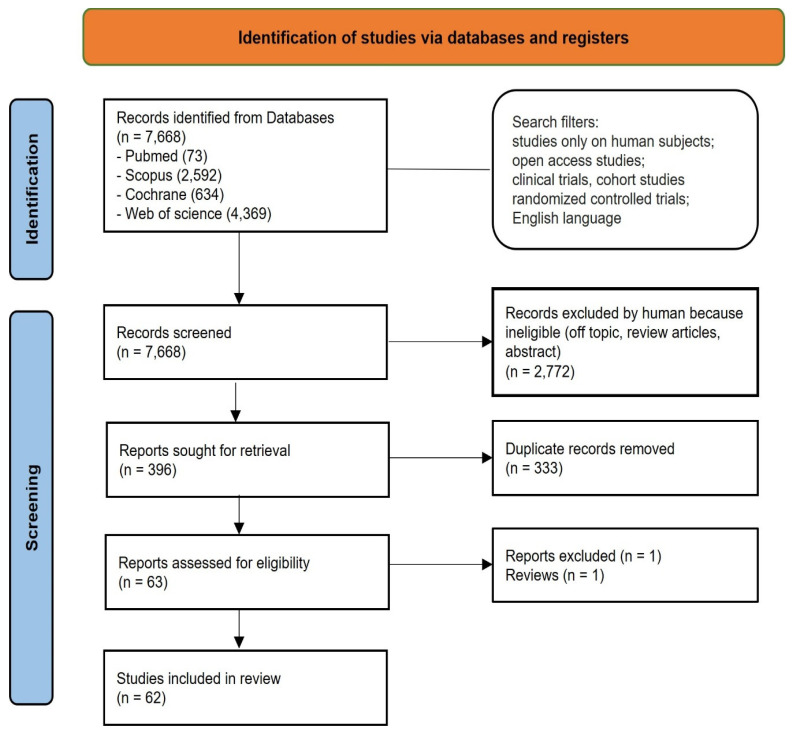
Literature search Preferred Reporting Items for Systematic Reviews and Meta-Analyses (PRISMA) flowchart diagram of the inclusion process.

**Figure 3 children-09-01846-f003:**
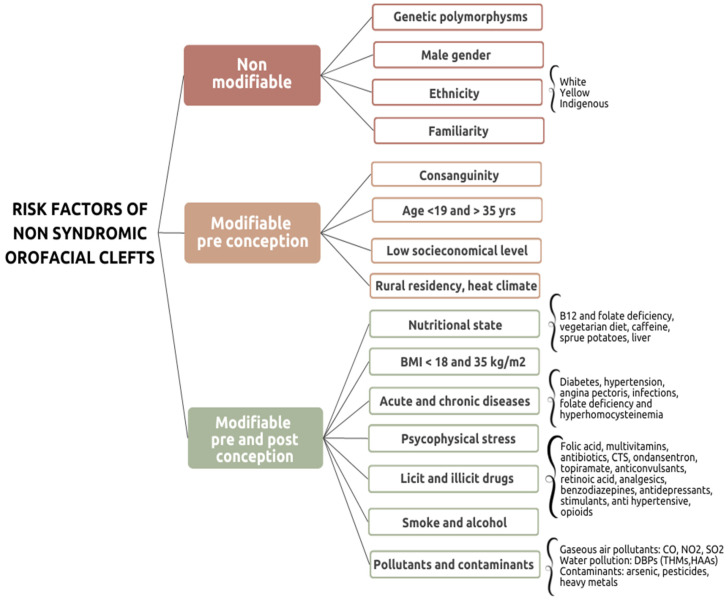
Overview of non-modifiable and modifiable risk factors of NSOFCs [CO: carbon monoxide; CTS: corticosteroids; DBPs: water disinfection By-products; HAAs: haloacetic acids; NO_2_: nitrogen dioxide; SO_2_: sulfur dioxide; THMs: trihalomethanes].

**Table 1 children-09-01846-t001:** Phenotypic description by LAHSHAL classification [CA: cleft alveolus; CL: cleft lip; CP: cleft palate; CL/P: cleft lip with or without palate].

LAHSAL ^1^ Abbreviated Notation	Phenotypic Description
[LAHS•••]	right unilateral complete CL, complete alveolus and complete “unilateral” CP
[laHS•••]	right unilateral incomplete CL and alveolus, complete “unilateral” CP
[•••SHal]	left unilateral incomplete CL and alveolus, with complete “unilateral” CP
[LAHSHAL]	bilateral symmetric complete CL, complete CA, and complete “bilateral” CP
[l*HSH*L]	bilateral symmetric incomplete CL, notched CA, and complete “bilateral” CP
[••HSH••]	complete “midline” cleft of hard and soft palate
[•••S•••]	complete “midline” cleft of soft palate

^1^ LAHSHAL is an acronym that indicates the anatomical structures, starting from the right side of the patient toward the left side: L (Right Lip); A (Right Alveolus); H (Right Hard Palate); S (Soft Palate); H (Hard Palate Sn); A (SN Alveolus); L (Sn Lip). The initial of the letter of the anatomical part involved is written uppercase or lowercase depending on whether the cleft is complete or partial. Minimal cleft is indicated by an asterisk (*); if the anatomical part is normally developed it is indicated by a dot (•); (e.g.,: laHS…: unilateral right incomplete cleft lip and alveolus, unilateral complete cleft palate). Based on these indication possibilities, the LAHSHAL system can describe over 12,000 combinations (of anatomy and severity) for CL/P.

## Data Availability

Not applicable.

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
