# Peer review of "Modifiable Risk Factors of Non-Syndromic Orofacial Clefts: A Systematic Review"

_children, 2022, doi:10.3390/children9121846_

Round 1
Reviewer 1 Report
Authors have conducted a systematic review to map potential risk factors for Non-Syndromic Orofacial Clefts. I found the systematic review soundly done following the guidlines. I cannot assess the clinical importance and relevance of the paper but I found the paper worthwhile in-terms of the methodological aspect the authors mainly focused on being systematic review for finding risk factors.
Best regards,
Author Response
All the authors are pleasured by the positive opinion and thank you.
Reviewer 2 Report
Thank You for having an opportunity to review this manuscript. I have to confess, that systematic reviews are not my favorit scientific articles as a whole.
But there are exceptions to the rules always. I feel this systematic review is one of these exceptions.
I liked the manuscript . It is structured, well organized and easy to read and summarising almost all relevant results regarding the topic: Etiology and risk factors of NSOFCs.
The methods, used , meets the requirements of a review article, the references are relevant and abundant ( 147 !). The statements and conclusions - although a number of them are ( and remain) debatable - are appropriate.
I have few remarks/ suggestions , mainly minor ones :
1. First sentence of the abstract states: " OFCs are the most frequent congenital defects, ....." I would suggest : " among the most frequent ", since hypospadias and club foot are equally frequent congenital anomalies.
2. Also in the abstract ( and later in the Results and in the Conclusions ) authors categorize the risk factors : " Risk factors were categorize into modifiable and non- modifiable . " but in the following sentences the sequence of the 2 groups, detailed ( i.e " first group" , "second group") is reverse in all 3 cases . The "non- modifiable " risk factors are listed as " first group" and the "modifiable" risk as the " second group".
3. A little bit more significant remark : In the Introduction Fig 1 mistakenly listed as fig 2. It showes a very complicated anatomical classification of OFCs with some drawings and a small table within the Fig. I suggest not to use this Fig for 2 resons: 1. very complicated , not easy to understand and interpret 2. Table 1 - LASHAL classification- that is also in the Introduction is much more appropriate and containes the necessary informations regarding the classification , Fig. 1 is unnecessary , in my opinion.
4. On page 15. two paragraphs ( 647-655 row) are discussed under Smoke and Alcohol ( 4.3.7 Chapter ) . However, these two paragraphs discuss ( licit) drugs, therefore should be listed in Chapter 4.3.6.
Apart from these remarks I suggest to accept the manuscript after minor revisions and to publish it in the Journal.
May I congratulate to those authors, who did most of the work.
Sincerely Yours:
Attila M Vastyan, MD, PhD, Med. Habil.
tenured professor in paediatric surgery
Pecs Cleft Team
University of Pecs
Pecs
Hungary
